# Structural shifts in food basket composition of rural and urban Philippines: Implications for the food supply system

**Subir Bairagi**[1]*, **Yacob Zereyesus**[2], **Sampriti Baruah**[3], **Samarendu Mohanty**[3]

**1** Department of Agricultural Economics and Agribusiness, University of Arkansas, Fayetteville, Arkansas, United States of America, **2** International Trade and Development Branch of the Market and Trade Economics Division, USDA Economic Research Service (ERS), Kansas City, Missouri, United States of America, **3** International Potato Center (CIP), New Delhi, India

* skbairag@uark.edu

**Data Availability Statement:** Data cannot be shared publicly because of Institutional restriction. However, the data are available from the Philippines Statistical Institutional Data Access for

## Abstract

Empirical evidence on the responsiveness and sensitivities of food consumption to its drivers is vital for conducting economic studies. Despite recent attempts to provide such estimates, much empirical work remains to be done considering the prevailing shifts in consumption trends in the Philippines. Price and expenditure elasticities are estimated for seven food categories for rural and urban Filipino households, using Stone–Lewbel (SL) price indices and the quadratic almost-ideal demand system (QUAIDS) model. We used multiple years (2006, 2009, 2012, 2015, and 2018) of the Philippines Family Income and Expenditure Survey (FIES) to estimate the food demand system. The results show that rice is a normal good for most households, particularly for rural consumers. However, it is an inferior good for the top 30% of rural Filipinos and the top 40% of urban Filipinos. As income increases, such wealthy households tend to replace their rice-dominated diet with nutrient-dense food products. Female-headed households, younger households, and households with educated members consume significantly more animal proteins such as meat and dairy products.

## Introduction

One-fifth and one-tenth of the total population in the Philippines are considered poor and food insecure, respectively [1, 2]. However, the Philippines has been one of the fastest-growing economies in Asia, with a current gross domestic product (GDP) of US$322 billion (2010 = 100) per annum. Over the past decade, its economy has recorded strong growth, more than 6% per year. Per capita income (US$3,022) and household consumption expenditures in the Philippines have also increased significantly [3]. There has also been a noticeable growing middle class and a large young population shaping the Philippines' economic dynamism [3]. Additionally, the burgeoning manufacturing and services sectors have attracted rural workers to urban areas by offering higher wages (income). Hence, an increasing rural out-migration is observed–the levels of urbanization changed from 45.3% to 51.2% from 2010 to 2015 [4].

researchers who meet the criteria for access to confidential data. Data can be accessed by contacting the Philippine Statistics Authority, Knowledge Management and Communications Division, Office of the National Statistician Email: information.center.psa@gmail.com Tel. No. 632-84626600 loc. 834/839 website: www.psa.gov.ph

**Funding:** We are grateful for the funding provided by the CGIAR Research Program on Climate Change, Agriculture and Food Security (CCAFS). The funders had no role in the study design, data collection and analysis, decision to publish, or preparation of the manuscript.

**Competing interests:** The authors have declared that no competing interests exist.

These ongoing structural changes, including population demographics, income, and urbanization, could be the primary drivers of diversified food demand in the Philippines. Similar changes in the food basket are also observed in many other developing countries worldwide, such as Bangladesh, China, India, and Vietnam. People from these countries are moving away from cereals to more high-value food consumption due to an increase in income and urbanization [5–10].

Research on understanding the evolution of food consumption patterns across rural and urban landscapes and among income groups in Asia has been growing. For example, for affluent urban households in Vietnam, the staple food, rice, became an inferior good and food preferences are evolving toward animal proteins (fish, pork, chicken, eggs, and milk), irrespective of urban status and income status [11]. As a result, Vietnamese urban households and high-income households are likely to put massive pressure on the country's food supply chain [12]. In Bangladesh, food preferences in urban areas are also evolving and future urban consumers are likely to eat less rice, the staple food, and consume more pulses and fish [10]. A similar pattern is also noted in China, where food budget shares of grains are declining, whereas foods with animal origins and fruits are increasing [13, 14]. The above studies applied the quadratic almost-ideal demand system (QUAIDS) model to estimate the own price and expenditure elasticities for food items. Some important limitations of these past studies are worth revisiting. For example, [10] estimated only five major food items, and hence the substitution effects for the remaining food items remain unknown. [12] used only a single year's data, so the drivers that affect food preferences over time are unknown. Although [13] used panel data, they did not consider estimating the elasticities of cereals at the disaggregated level, rice versus other cereals [13]. Thus, the diversification of the preferences within cereal consumption could not be observed. Finally, because none of these studies mentioned above used actual market prices of food items to estimate the demand elasticities, estimation biases might arise as this was not controlled for the quality issue [15–17]. A detailed review of previous demand studies can be found in [11].

The Philippines government uses various domestic and trade policies, such as high import tariffs and production subsidies in rice, the staple food in the country, to enhance food security for marginalized people [18]. To meet the demand for high-value foods, it is also equally important to improve our understanding of the demand-side drivers in the face of changing food preferences [19–21], besides understanding the supply-side drivers (e.g., trade-offs between agricultural productivity and environmental sustainability) [21]. Food consumption patterns could also evolve differently between rural and urban households and different income groups [5, 10–12, 21–23]. Considering these factors, disaggregated food demand analysis by urban and rural households and different income groups is fundamental for accurately analyzing the food and nutrition trends in any country.

The purpose of the current study is to examine the drivers of food demand and to estimate the price and expenditure elasticities for seven food items (rice, other cereals, meat, fish, dairy products, fruits and vegetables, and miscellaneous food) for rural and urban Filipino households. To do so, the study applied the Stone–Lewbel (SL) price indices and the quadratic almost-ideal demand system (QUAIDS) model using multiple years (2006, 2009, 2012, 2015, and 2018) of the Philippines Family Income and Expenditure Survey (FIES) to estimate the food demand system.

The present study contributes to the food demand literature in multiple ways. First, to the best of our knowledge, no study has been conducted in the Philippines to understand the evolution of consumption patterns of rural and urban households by estimating a complete food demand system, except for two relatively old studies of food demand analysis [24, 25]. Second, the analysis is disaggregated by household income and urban status. Third, a richer dataset of

five years of income and expenditure surveys is used for the analysis, allowing an estimation of the evolution of the food demand system and changing preferences over time. Fourth, estimation is disaggregated for rice and other cereals to account for the implications of the intervention by the Philippines' government in the rice sector. Finally, income and regional dummies are used as instruments for expenditure to control the endogeneity problem in the estimation of the QUAIDS models.

The rest of the article is organized as follows. The next section presents a brief literature review on the evolution of food demand in Asia. This is followed by a data and methods section elaborating on the descriptive statistics, models, and empirical strategy used for estimations. The results section presents the main findings from the study and discussion in the context of other relevant studies. The final section concludes the manuscript with recommendations and possible future extensions of the study.

## Evolution of food demand

Many developing countries have been growing at a much faster rate than the developed countries, which has significant ramifications for global food demand [26]. A growing body of empirical literature has noted that the food basket is evolving, shifting away from starchy staples toward animal-based products and fruits and vegetables [10, 11, 26–30]. Therefore, supplying this increased diversified food demand will put tremendous pressure on agricultural resources in the future. However, there is considerable heterogeneity in the convergence of food baskets across developing countries, such as cereals to animal-based versus plant-based proteins. Food preference convergences could also vary according to rural and urban status and income status [5, 10–12, 21–23]. Therefore, country-specific demand studies are needed to improve our understanding of the demand-side drivers in the face of changing food preferences, which will help guide the framing of better food policies for a country [19–21].

There is a long history of the evolution of food demand studies. For example, Deaton's contributions helped extend such work in multiple dimensions, including studying the relationship between the consumption of goods and services and human welfare, which is crucial for designing economic policy [15, 31–33]. Several systematic research reviews on food demand (meta-analysis) are abundant in the recent literature [34–37]. Additionally, a rigorous review of the literature on food demand in Asia can be found in Bairagi et al. [11]. Therefore, we briefly discuss here the literature on recent food demand studies.

Food demand studies have policy implications for optimally allocating agricultural resources to meet food demand, particularly in poor and developing economies. For example, policymakers may be able to design policies that improve food and nutrition security and alleviate poverty based on how price and income changes affect demand for various foods, measured by price and income elasticities, respectively. For example, price elasticity of demand (PE) measures how consumers' demand changes as price changes. PE values are measured as the ratio of the percentage change in demand associated with the percentage change in the price of that good (own-PE) or another good (cross-PE). The sign of own-PE is almost always negative, but the sign of cross-PE could be positive or negative depending on the substitutability or complementarity nature of the good [38]. For a necessity good, demand is inelastic (PE <1), meaning that demand does not respond much to price changes, whereas demand changes by a greater proportion when the price changes (i.e. PE >1) is referred to as a luxury good. The sizes of price elasticities vary widely based on whether a country is poor or developed [36, 37]. For example, based on 78 global studies, Cornelsen et al. [37] and Green et al. [39] found that the own-PE for cereals in low-, medium-, and high-income countries were −0.61, −0.55, and −0.43, respectively. For meat, the own-PE in low-, medium-, and high-income countries

were −0.78, −0.72, and 0.60, respectively. Price elasticities also vary based on food types. For instance, for rice, one of the main staple foods in Asia, the own-PE ranged from −0.04 to −2.17 [10, 12, 16, 17, 40]. This considerable variation is attributable to various reasons, including the time and the country considered in the analysis, the quality of the data used, and the robustness of the method applied.

Expenditure (income) elasticity measures how consumers' food demand changes (on either a quantity or caloric basis) as their income changes. The relationship between food consumption and income, known as Engel's curve, is considered positive and linear in most instances. However, food demand literature that shows a non-linear relationship is also plausible because consumers might spend in various ways when their income increases [35, 41]. For instance, consumers may spend more on rice and wheat when their income increases slightly. However, the consumption of meat and dairy products may increase with consumers' affluent stage [42]. More importantly, this also depends on various factors, such as food choices, income groups, and location. Therefore, expenditure elasticities also diverge widely, as is the case with price elasticities. Bairagi et al. [11] noted a wide range of income elasticities, ranging from −0.50 to 3.54, based on more than 33 previous studies in Asia. The primary factors of this substantial variation are food items studied, publication bias, and methodological attributes (e.g., model type, sample size, types of data) [34, 35]. However, most studies concur that income growth will increase food consumption and lead to more nutritionally diverse diets.

In summary, elasticities are essential policy tools for guiding price and food policies. Demand studies are needed as new information becomes available. Second, food preferences vary widely, and there are considerable variations in the estimated price and income elasticities across countries [15]. Third, a robust approach is needed to minimize the potential estimation biases in estimated elasticities due to methodological and data quality issues [15]. Finally, updated cross-PEs estimates, limited in the demand literature [37], are crucial with recent data to help draw evidence-based policies for further economic development in many developing countries [11].

## Methods

### Stone–Lewbel (SL) price indices

As mentioned, the FIES does not collect the market prices of food items at the household level. Since consumer price indices (CPIs) are available at the regional level, they can be used for demand system estimation. However, using such prices may lead to insufficient variation within observations, resulting in biased estimations of consumer demand models. As a remedy, Stone–Lewbel (SL) price indices are used to estimate commodity price indices at the household level using the information on the budget shares and the CPIs of the goods comprising the commodity groups [43]. To estimate SL price indices, assumptions are made that the between-group utility function is weakly separable, and the within-group utility functions are of the Cobb–Douglas form. Therefore, "individual-specific price indices allow for a population with heterogeneity in preferences for goods within a given bundle of goods" [44]. Empirical evidence shows that it is possible to accurately estimate a demand system based on SL price indices, and elasticities and marginal effect estimates are robust [44–46].

Let $n$ be the number of groups and $n_i$ be the number of food items in group $i$, where $i = 1, \ldots, N$, and in our case, $N = 7$; $q_{ij}$ and $p_{ij}$, respectively, denote the quantities and the prices of the j-th food items, where $j = 1, \ldots n$. Let $x$ be total expenditures and $x_i$ be total expenditures in group $i$. Then, the budget share of group $i$ is $w_i = x_i/x$ and the within-group budget share of the j-th item in group $i$ is $w_{ij} = p_{ij}q_{ij}/x_i$. Let $s$ be a vector of observed demographic

characteristics. Also, assume that $U(u_1(q_1, s),\ldots,u_n(q_n, s))$ is a weakly separable utility function, where $U(u_1,\ldots,u_n)$ is the between-group utility function and $u_i(q_i, s)$ is the within-group sub-utility function for group $i$. This within-group utility function is assumed to be the Cobb–Douglas form, expressed as $u_i(q_i, s) = k_i \prod_{j=1}^{n_i} q_{ij}^{w_{ij}}$, where $k$ is a scaling factor that takes the following form: $k_i = \prod_{j=1}^{n_i} \bar{w}_{ij}^{-\bar{w}_{ij}}$, where $\bar{w}_{ij}$ is the budget share of good $j$ in group $i$ of the reference household. Then, the SL price index can be estimated as

$$v_i(p_i, s) = \frac{1}{k_i} \prod_{j=1}^{n_i} \left(\frac{p_{ij}}{w_{ij}}\right)^{w_{ij}} \tag{1}$$

Eq (1) represents the product of three components: (i) a scaling factor, $k$, which is invariant across households. In this regard, the reference household considered is the one whose budget share is equivalent to the sample mean of budget share (the regional mean budget share is considered in our case); (ii) a commodity group price component, $p_{ij}$, for which the annual regional CPIs (2012 = 100) are used that vary across households; and (iii) finally, a commodity group budget share component, $w_{ij}$, which is specific to each household. The main goal of estimating SL price indices is to have enough variability in price indices since CPIs are not disaggregated enough across commodities. The main source of variability comes from sub-group budget shares. However, if there are few sub-groups under a specific group and the share of one of the sub-groups is large, there might not be enough variability in the SL price index. The standard deviations (within parentheses) presented in Table 1 reveal that the estimated SL price indices have enough variability. Moreover, prices of all commodities increased over time, consistent with the commodity price data in the Philippines.

## Specification of food demand systems

To estimate price and expenditure elasticities for the seven food items in the Philippines, we used the quadratic almost-ideal demand system (QUAIDS), which is a variant of the AIDS model [31, 32, 47]. The estimation is further augmented with demographic variables and with instrumental variables. Recently, the QUAIDS has been widely applied in the literature for food demand analysis because of the computational advantages of estimating large demand systems through the development of advanced statistical software [10–14, 27, 48–51]. Hence, we briefly discuss the basic demand model below.

**Table 1. Estimated Stone–Lewbel (SL) price indices.**

| Commodity groups | Rural | | | | | | Urban | | | | | |
|---|---|---|---|---|---|---|---|---|---|---|---|---|
| | 2006 | 2009 | 2012 | 2015 | 2018 | Average | 2006 | 2009 | 2012 | 2015 | 2018 | Average |
| Price of rice | 56 (13) | 72 (19) | 85 (19) | 96 (23) | 108 (25) | 93 (29) | 57 (11) | 72 (19) | 86 (17) | 96 (23) | 108 (23) | 93 (27) |
| Price of maize and other cereals | 60 (14) | 78 (18) | 91 (18) | 98 (20) | 106 (21) | 94 (25) | 61 (13) | 78 (16) | 89 (18) | 96 (20) | 104 (21) | 92 (25) |
| Price of meat | 61 (16) | 74 (18) | 83 (18) | 92 (19) | 104 (19) | 91 (24) | 68 (13) | 82 (15) | 88 (16) | 94 (17) | 107 (18) | 95 (22) |
| Price of fish | 66 (11) | 82 (13) | 92 (16) | 102 (19) | 121 (23) | 103 (27) | 67 (13) | 81 (14) | 90 (18) | 98 (20) | 122 (25) | 103 (30) |
| Price of dairy products | 67 (11) | 81 (13) | 97 (12) | 105 (13) | 113 (13) | 100 (20) | 68 (10) | 82 (12) | 96 (13) | 104 (14) | 112 (13) | 100 (20) |
| Price of fruits and vegetables | 64 (8) | 81 (10) | 95 (10) | 110 (13) | 135 (15) | 111 (29) | 68 (8) | 83 (9) | 96 (10) | 110 (12) | 138 (15) | 113 (30) |
| Price of miscellaneous food items | 70 (9) | 84 (10) | 89 (10) | 94 (13) | 106 (18) | 95 (19) | 70 (11) | 84 (13) | 88 (11) | 91 (14) | 108 (19) | 95 (21) |

Notes: Numbers in parentheses are standard deviations.

Suppose there are $N$ budget (expenditure) share equations for food items $i$ for household $h$ (= $1, \ldots, H$). The system of food demand equations can be expressed as

$$w_i^h = \alpha_i + \gamma_i' \boldsymbol{p}^h + \beta_i \{ x^h - a(\boldsymbol{p}^h, \theta) \} + \lambda_i \frac{\{ x^h - a(\boldsymbol{p}^h, \theta) \}^2}{b(\boldsymbol{p}^h, \theta)} + u_i^h, \tag{2}$$

along with the following non-linear price aggregators, $a(\boldsymbol{p}^h, \theta) = \alpha_0 + \alpha' \boldsymbol{p}^h + \frac{1}{2} \boldsymbol{p}^{h'} \Gamma \boldsymbol{p}^h$ and $b(\boldsymbol{p}^h, \theta) = \exp(\beta' \boldsymbol{p}^h)$, where $x^h$ is the log of total expenditures; $\boldsymbol{p}^h$ is the vector of prices of $N$ food items (in our case, the estimated SL price indices); $\alpha = (\alpha_1, \ldots, \alpha_N)'$, $\beta = (\beta_1, \ldots, \beta_N)'$, $\Gamma = (\gamma_1, \ldots, \gamma_N)'$, and $\theta$ are the set of all parameters to be estimated; and $u_i^h$ is an error term. All of these parameters must satisfy the theoretical adding up, homogeneity, and symmetry restrictions. Adding up implies that the budget shares add up to 1 and all parameters must sum to zero over all equations except the constant term; homogeneity implies that the log price parameters must sum to zero within each equation; symmetry implies that the effect of log price $i$ on budget share $j$ must equal the effect of log price $j$ on budget share $i$.

Differentiating Eq (2) with respect to $x$ and $p_j$, omitting $h$ superscripts, will give the following equations:

$$u_i = \beta_i + 2\lambda_i \frac{\{ x - a(\boldsymbol{p}, \theta) \}}{b(\boldsymbol{p}, \theta)}, \tag{3}$$

and

$$u_{ij} = \lambda_{ij} + u_i \left( \alpha_j + \gamma_j \boldsymbol{p} \right) - \lambda_i \beta_j \frac{\{ x - a(\boldsymbol{p}, \theta) \}^2}{b(\boldsymbol{p}, \theta)}. \tag{4}$$

Price and expenditure elasticities can be computed from Eqs (3–4) as (i) expenditure elasticities: $e_i = \frac{u_i}{w_i} + 1$; (ii) uncompensated price elasticities: $e_{ij}^u = \frac{u_{ij}}{w_i} - \delta_{ij}$, where $\delta_{ij}$ is the Kronecker delta; and (iii) compensated price elasticities: $e_{ij}^c = e_{ij}^u + e_i w_j$.

To account for household heterogeneities, Poi [52] extended Eq (2) with household demographic variables, following a method proposed by [53] called the translating approach. This approach allows the level of demand to be dependent upon demographic variables, which were included through the constant term in Eq (2) as $\alpha^h = A s^h$, where $A = \alpha_i'$, a linear combination of a set of demographic variables $s^h$. In this study, we used the following six demographic variables for controlling household-level heterogeneities: gender, age, education, and marital status of the HHs; numbers of economically active family members; and spouse employment status. Finally, to account for endogeneity of prices and total expenditure variables in Eq (2), we used the instrumental variables (IV) method [54], where log of income and regional dummies are used as instruments to augment with the predicted error vector $\hat{\boldsymbol{v}}^h$ from estimating reduced forms of $x^h$ and $\boldsymbol{p}^h$. The error term can be written via the orthogonal decomposition, $u_i^h = \boldsymbol{\rho}_i \hat{\boldsymbol{v}}^h + \varepsilon_i^h$, along with assuming $E(\varepsilon_i^h | x^h, \boldsymbol{p}^h) = 0$ for all $i$ and $h$. For estimating the demand system, we used the written STATA codes [52, 55]. We also set $\alpha_0 = 10$, slightly less than the mean value of the log of expenditure [32].

## Data

We used the Philippines Family Income and Expenditure Survey (FIES) from multiple years (2006, 2009, 2012, 2015, and 2018) to estimate the demand system. The FIES is a nationwide survey of households undertaken by the Philippine Statistics Authority (PSA) that collects detailed information on family income and expenditures every three years. Thus far, the PSA has conducted 18 surveys since 1959. In most of the recent surveys, about 40,000 sample

households were interviewed in each survey. Note that, for the first time, the PSA covered nearly 150,000 households in the 2018 FIES. These households were selected from the country's 17 administrative regions (National Capital Region, Cordillera Administrative Region, Ilocos, Cagayan Valley, Central Luzon, Calabarzon, Mimaropa, Bicol, Western Visayas, Central Visayas, Eastern Visayas, Zamboanga Peninsula, Northern Mindanao, Davao, Soccsksargen, Caraga, and Autonomous Region in Muslim Mindanao). To collect the data from the household heads (HHs), the PSA used a stratified random sampling technique. The details of the sampling design and data collection technique can be found in the 2018 FIES report [56].

The FIES questionnaires cover a wide range of areas, such as household characteristics, household income and expenditures, social protection, and entrepreneurial activities. For this study, the main variables of interest are household income and expenditures on different food items and demographic characteristics of the HHs. The PSA first divided food expenditures into two broad categories, food consumed at home and food consumed outside the home. Food consumed at home was then divided into 12 sub-groups [56]. However, we have regrouped it into seven major categories (rice, other cereals, meat, fish, dairy products, fruits and vegetables, and miscellaneous items) because of the unavailability of market prices at the disaggregated level (Fig 1 and Table 3). Importantly, the PSA collects information on household expenditures using the Philippine peso (PHP) on these commodities, but not market prices (unit prices). Therefore, we have estimated Stone–Lewbel (SL) price indices for the food items at the household level using regional consumer price indices and sub-group budget shares (details appear in the methodology section).

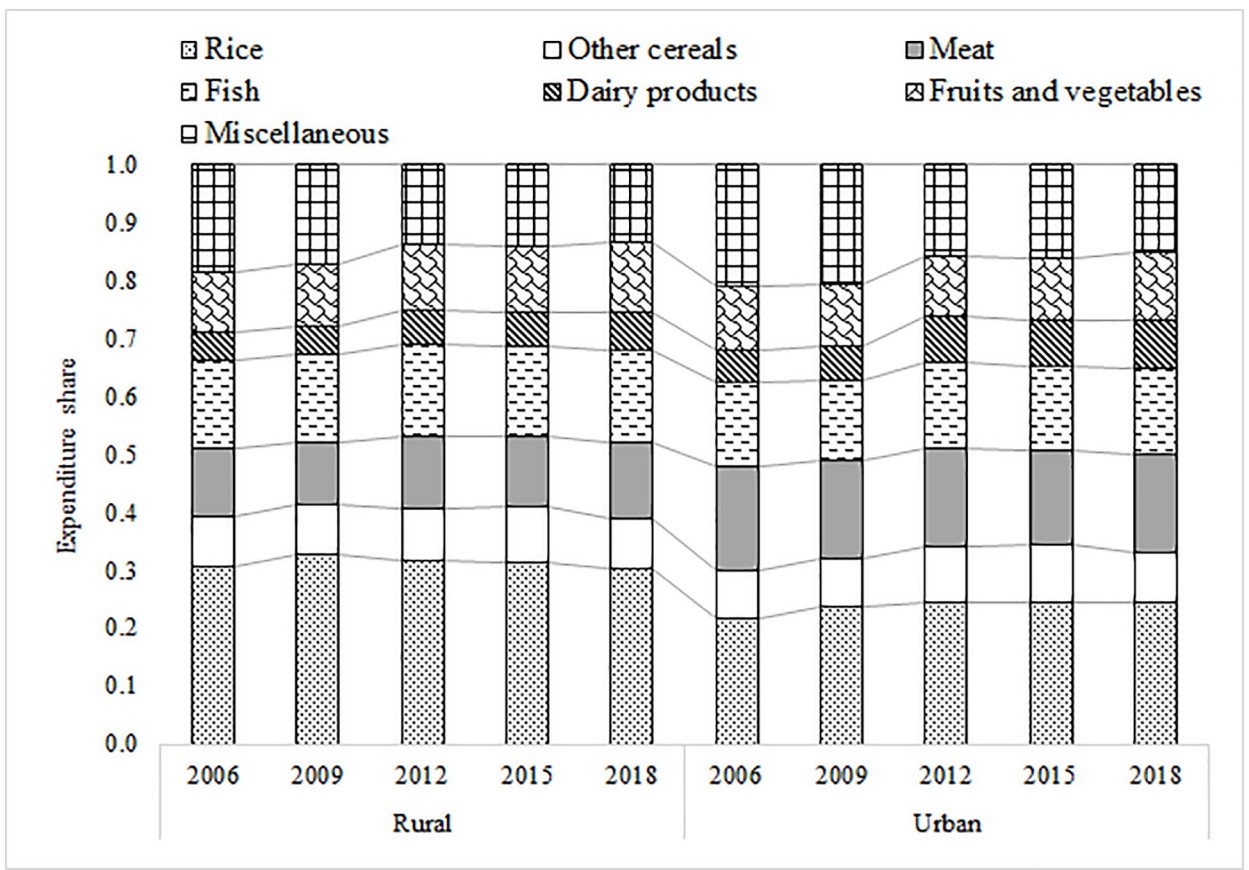

**Fig 1. Evolution of food composition patterns in the Philippines (expenditure shares across rural-urban landscape, 2006–2018).** Source: Authors' estimation based on FIES data.

**Table 2. Household sociodemographic characteristics: 2016–2018.**

| Variables | Rural | | | | | | Urban | | | | | | Mean difference (*t*-test) |
|---|---|---|---|---|---|---|---|---|---|---|---|---|---|
| | 2006 | 2009 | 2012 | 2015 | 2018 | All | 2006 | 2009 | 2012 | 2015 | 2018 | All | |
| *Demographic characteristics* | | | | | | | | | | | | | |
| Gender of HH (male = 1) | 0.85 | 0.83 | 0.81 | 0.81 | 0.80 | 0.81 | 0.79 | 0.76 | 0.74 | 0.74 | 0.74 | 0.75 | 0.063*** |
| Age of HH (years) | 48.73 | 50.39 | 51.20 | 51.60 | 50.69 | 50.63 | 48.05 | 49.59 | 50.02 | 50.87 | 49.60 | 49.59 | 1.033*** |
| Marital status of HH (married = 1) | 0.82 | 0.80 | 0.78 | 0.77 | 0.78 | 0.78 | 0.79 | 0.77 | 0.74 | 0.73 | 0.75 | 0.75 | 0.030*** |
| HH had no schooling (yes = 1) | 0.05 | 0.05 | 0.04 | 0.04 | 0.03 | 0.04 | 0.01 | 0.01 | 0.01 | 0.01 | 0.01 | 0.01 | 0.027*** |
| HH had elementary education (yes = 1) | 0.53 | 0.52 | 0.49 | 0.48 | 0.80 | 0.64 | 0.28 | 0.28 | 0.28 | 0.25 | 0.68 | 0.48 | 0.163*** |
| HH had high school education (yes = 1) | 0.29 | 0.30 | 0.33 | 0.33 | 0.02 | 0.18 | 0.39 | 0.39 | 0.40 | 0.41 | 0.03 | 0.21 | -0.035*** |
| HH had college undergrad education and specialized training (yes = 1) | 0.07 | 0.07 | 0.09 | 0.09 | 0.15 | 0.11 | 0.16 | 0.16 | 0.18 | 0.20 | 0.27 | 0.23 | -0.114*** |
| HH had college and above level of education (yes = 1) | 0.05 | 0.05 | 0.06 | 0.06 | 0.00 | 0.03 | 0.15 | 0.16 | 0.13 | 0.13 | 0.00 | 0.07 | -0.041*** |
| Economically active family members (above 17 years old) | 3.76 | 3.81 | 2.89 | 2.93 | 2.90 | 3.12 | 3.98 | 3.97 | 3.06 | 3.08 | 3.08 | 3.31 | -0.191*** |
| Employment status of HH's spouse (employed = 1) | 0.40 | 0.40 | 0.43 | 0.43 | 0.42 | 0.42 | 0.40 | 0.40 | 0.40 | 0.39 | 0.40 | 0.40 | 0.022*** |
| *Economic characteristics* | | | | | | | | | | | | | |
| Real per capita income (PHP/year) | 33626 | 35678 | 40432 | 43699 | 48394 | 43246 | 65879 | 67539 | 68523 | 74307 | 75616 | 72316 | -55668*** |
| Real per capita expenditure (PHP/year) | 29351 | 31016 | 32782 | 34918 | 36401 | 34165 | 55996 | 58240 | 57169 | 60984 | 58710 | 58381 | -44513*** |
| Food expenditure share | 0.54 | 0.55 | 0.56 | 0.53 | 0.53 | 0.54 | 0.46 | 0.47 | 0.47 | 0.46 | 0.47 | 0.47 | 0.074*** |
| Food consumed outside household | 0.06 | 0.06 | 0.09 | 0.10 | 0.10 | 0.09 | 0.13 | 0.14 | 0.18 | 0.21 | 0.20 | 0.18 | -0.092*** |
| Selection bias (if a household did not consume any food item, yes = 1) | 0.22 | 0.25 | 0.03 | 0.03 | 0.01 | 0.07 | 0.14 | 0.15 | 0.02 | 0.02 | 0.01 | 0.05 | 0.025*** |
| Sample size | 20,463 | 20,349 | 23,901 | 25,210 | 79,005 | 168,928 | 16,373 | 16,370 | 14,813 | 14,716 | 63,787 | 126,059 | |

Source: Authors' computation based on FIES data. PHP denotes the Philippine peso (USD 1.00 = PHP 52.00). HH = household head. The mean difference is estimated for all samples.

Table 2 illustrates the sociodemographic characteristics of the HHs in rural and urban areas. In the last column of Table 2, the mean differences of sociodemographic variables by urban status indicate that urban households are significantly different from rural households. For instance, younger and more highly educated households live in urban areas, which is a major demographic shift that is hypothesized to affect food demand structure. Approximately 30% of the urban HHs have college and above education levels, compared to only 14% for their rural counterparts. The urban households have more economically active family members, which may explain the higher family income than that of their rural counterparts. The per capita income and expenditures of urban households are found to be nearly double those of their rural counterparts. Budget shares show that the urban households spend proportionally less on food consumption at home than their rural counterparts (47% versus 54%), but urban households spend proportionally twice as much on eating food outside of their home (18% versus 9%). Table 2 further reveals that per capita income in both rural and urban households increased considerably during 2006–2018, almost at the same rate. While the per capita food expenditures have increased for rural households, such expenditures for urban households remained somewhat constant until 2015 and then slightly decreased. Finally, food expenditure shares for both rural and urban households have remained constant during the past decade.

Fig 1 and Table 3 present the food basket composition in the Philippines during 2006–2018. As mentioned, the Filipino food basket (food consumed at home) contains seven major

Table 3. Composition of the food basket in the Philippines.

| Composition of food groups and sub-groups | Budget share (%) by year | | | | | | Budget share (%) by region | | |
|---|---|---|---|---|---|---|---|---|---|
| | 2006 | 2009 | 2012 | 2015 | 2018 | Average (2006–2018) | Rural | Urban | Rural-urban differences (*t*-test) |
| (1) **Rice** (well-milled rice, regular and NFA rice) | 0.266 | 0.288 | 0.289 | 0.288 | 0.275 | 0.279 | 0.309 | 0.239 | 0.072*** |
| (2) **Maize and other cereals** (maize, flour, cereal preparation, bread, pasta and other bakery products) | 0.085 | 0.082 | 0.093 | 0.097 | 0.089 | 0.089 | 0.089 | 0.089 | -0.000*** |
| (3) **Meat** (pork, chicken, beef, and preserved meat) | 0.144 | 0.136 | 0.140 | 0.137 | 0.148 | 0.144 | 0.124 | 0.170 | -0.045*** |
| (4) **Fish** (fresh fish, seafood (e.g., shrimp, crab, squid, and shell) and dried and preserved fish and seafood) | 0.148 | 0.145 | 0.156 | 0.150 | 0.152 | 0.151 | 0.155 | 0.146 | 0.009*** |
| (5) **Dairy products** (eggs and milk and milk products) | 0.052 | 0.053 | 0.066 | 0.068 | 0.075 | 0.067 | 0.061 | 0.076 | -0.015*** |
| (6) **Fruits and vegetables** | 0.107 | 0.105 | 0.110 | 0.110 | 0.118 | 0.113 | 0.115 | 0.110 | 0.004*** |
| (7) **Miscellaneous** (edible oils, sugar, jam, honey, non-alcoholic beverages, coffee, cocoa, tea, and food products not elsewhere classified) | 0.198 | 0.190 | 0.145 | 0.150 | 0.143 | 0.157 | 0.147 | 0.170 | -0.023*** |

Notes: Authors' computation based on FIES data. NFA stands for National Food Authority. The detailed sub-commodity group shares are presented in S1 and S2 Tables.

food items: rice, other cereals, meat, fish, dairy products, fruits and vegetables, and miscellaneous items. The results indicate that, on average, a Filipino spent nearly two-thirds of the total food budget on consuming the following three food items: rice, meat, and fish. The results further indicate three major structural shifts within the Filipino food basket: (i) First, the budget share of the leading staple food (rice) is the highest in both rural and urban areas. However, on average, rural residents spent more on rice (almost six percentage points higher) than their urban counterparts. For cereal (rice, maize, and other cereals) consumption, budget shares remained constant during the past decade. This could be for various reasons, including consumer preferences toward premium-quality rice and/or substitutes of rice consumption. (ii) Second, spending on dairy consumption (eggs and milk and milk products) is on the rise for rural and urban residents. (iii) Finally, urban residents spent a significantly higher proportion of their food budget on meat and dairy products than rural residents during 2006–2018 (5 and 2 percentage points higher, respectively) (Table 3).

## Results and discussion

### Results from reduced form instrumental regression

The estimated parameters from the reduced form expenditure equations, using income and regional dummy variables as instruments, are presented in Table 4, with the following five main insights. (i) The instrumental variable, income, is positively and significantly associated with expenditure, implying a valid instrument. The regional dummy variables affect food expenditure differently (positively/negatively), implying the presence of food choice heterogeneities among the regions. (ii) Commodity prices are negatively and significantly associated with food expenditure, except for the prices of meat, fruits and vegetables, and miscellaneous foods. This indicates that, since the consumer budget is fixed, consumers respond to changes in commodity prices and adjust their food choices accordingly. (iii) Education is positively associated with expenditure, indicating that households with highly educated members are likely to spend more than households with no education. (iv) Household expenditure increases if a household has a higher number of economically active family members and the spouse is employed. (v) Finally, female-headed households in both urban and rural areas spend more on food consumption than their male-headed counterparts. Compared with their older counterparts, a household headed by a younger person spends more on food.

**Table 4. Parameters estimated from instrumental regression (endogenous expenditure function).**

| Dependent variable: log of expenditure | Rural | Urban |
|---|---|---|
| Log of rice price | -0.054*** (0.004) | -0.053*** (0.006) |
| Log of other cereal prices | -0.046*** (0.005) | -0.050*** (0.006) |
| Log of meat price | 0.249*** (0.004) | 0.362*** (0.006) |
| Log of fish price | -0.127*** (0.005) | -0.102*** (0.005) |
| Log of dairy price | -0.199*** (0.006) | -0.404*** (0.007) |
| Log of fruits and vegetables price | 0.098*** (0.007) | 0.214*** (0.009) |
| Log of miscellaneous food price | 0.082*** (0.006) | 0.027*** (0.006) |
| Log of per capita income | 0.173*** (0.001) | 0.138*** (0.002) |
| Gender of HH (male = 1) | 0.006** (0.003) | -0.011*** (0.003) |
| Age of HH, log | -0.165*** (0.003) | -0.084*** (0.004) |
| Marital status of HH (married = 1) | 0.156*** (0.003) | 0.146*** (0.003) |
| Education level of HH (elementary school = 1) | 0.040*** (0.005) | 0.043*** (0.009) |
| Education level of HH (high school = 1) † | 0.086*** (0.005) | 0.090*** (0.010) |
| Education level of HH (below college and specialized training = 1) † | 0.133*** (0.005) | 0.106*** (0.010) |
| Education level of HH (college and above = 1) † | 0.173*** (0.007) | 0.165*** (0.010) |
| No. of economically active family members, log | 0.516*** (0.002) | 0.511*** (0.002) |
| Spouse employment (employed = 1) | 0.036*** (0.002) | 0.034*** (0.002) |
| Time index | 0.026*** (0.002) | 0.008*** (0.002) |
| Selection bias (if a household did not consume any food item, yes = 1) | 0.010*** (0.004) | -0.030*** (0.002) |
| Constant | 8.866*** (0.053) | 9.063*** (0.065) |
| Regional effects | Yes | Yes |
| Adjusted $R^2$ | 0.462 | 0.473 |
| Observations | 168,857 | 126,009 |

Notes

*, **, and *** denote statistical significance levels at 10, 5, and 1 percent, respectively. HH = household head.

† Base level is no education.

## Drivers of food demand in the Philippines

The parameter estimates from QUAIDS models for rural and urban households are revealed in Tables 5 and 6, respectively, which indicate the role of socioeconomic and demographic variables in food consumption in the Philippines. The results underline the importance of four major drivers of food demand. First, the coefficients of the time index for rural and urban households are negative and statistically significant for cereals (rice and other cereals) but positive for all other high-value food items. This means that Filipino consumers have been spending less and less on cereal consumption over time but more on other food items. This finding perhaps indicates that consumer preferences are shifting away from cereals. This food demand reorientation has implications for a diversified and sustainable food system, particularly reshaping more on the non-rice sector to supply highly nutritious food items at affordable prices to various groups in the Philippines.

Second is the emerging role of gender in food consumption. Female-headed households tend to spend more on nutrient-rich foods such as meat, dairy products, and fruits and vegetables than their male-headed counterparts in the Philippines. This finding is consistent with the fact that women in many developing countries are increasingly becoming household heads and decision-makers since they are participating more in the labor market and male family members are out-migrating for a higher income [57, 58]. A positive association between

**Table 5. Parameter estimates from QUAIDS model for rural Filipino households, 2006–2018.**

| Variables | Rice | Other cereals | Meat | Fish | Dairy products | Fruits and vegetables | Miscellaneous |
|---|---|---|---|---|---|---|---|
| Log of prices | 0.280*** (0.01) | -0.038*** (0.00) | -0.170*** (0.00) | 0.0056*** (0.00) | -0.059*** (0.00) | 0.0034** (0.00) | -0.019*** (0.00) |
| | | 0.023*** (0.00) | -0.0060*** (0.00) | -0.0028*** (0.00) | 0.0073*** (0.00) | 0.019*** (0.00) | -0.0024*** (0.00) |
| | | | 0.13*** (0.00) | -0.0096*** (0.00) | 0.063*** (0.00) | -0.018*** (0.00) | 0.015*** (0.00) |
| | | | | -0.089*** (0.00) | 0.045*** (0.00) | 0.015*** (0.00) | 0.035*** (0.00) |
| | | | | | -0.11*** (0.00) | 0.014*** (0.00) | 0.043*** (0.00) |
| | | | | | | -0.020*** (0.00) | -0.013*** (0.00) |
| | | | | | | | -0.059*** (0.00) |
| Log of expenditure | -0.34*** (0.00) | 0.014*** (0.00) | 0.21*** (0.00) | 0.0072*** (0.00) | 0.098*** (0.00) | -0.018*** (0.00) | 0.030*** (0.00) |
| Log of expenditure squared | -0.012*** (0.00) | 0.0078*** (0.00) | 0.0091*** (0.00) | -0.0017*** (0.00) | 0.0028*** (0.00) | -0.0027*** (0.00) | -0.0038*** (0.00) |
| Residuals of expenditure | 0.21*** (0.00) | 0.047*** (0.00) | -0.10*** (0.00) | -0.0048*** (0.00) | -0.067*** (0.00) | -0.013*** (0.00) | -0.069*** (0.00) |
| Gender of HH (male = 1) | 0.016*** (0.00) | -0.0056*** (0.00) | -0.0055*** (0.00) | 0.010*** (0.00) | -0.010*** (0.00) | -0.0031*** (0.00) | -0.0016*** (0.00) |
| Age of HH, log | -0.029*** (0.00) | -0.013*** (0.00) | 0.020*** (0.00) | 0.0059*** (0.00) | 0.0077*** (0.00) | 0.016*** (0.00) | -0.0082*** (0.00) |
| Marital status of HH (married = 1) | 0.035*** (0.00) | 0.012*** (0.00) | -0.023*** (0.00) | -0.0064*** (0.00) | -0.0074*** (0.00) | 0.0020*** (0.00) | -0.011*** (0.00) |
| Education level of HH (elementary school = 1) | 0.0029* (0.00) | 0.0037*** (0.00) | 0.021*** (0.00) | -0.0100*** (0.00) | 0.0041*** (0.00) | -0.015*** (0.00) | -0.0066*** (0.00) |
| Education level of HH (high school = 1) [†] | 0.0031 (0.00) | 0.0040*** (0.00) | 0.029*** (0.00) | -0.018*** (0.00) | 0.0069*** (0.00) | -0.016*** (0.00) | -0.0090*** (0.00) |
| Education level of HH (below college and other specialized training = 1) [†] | -0.0033 (0.00) | 0.0085*** (0.00) | 0.030*** (0.00) | -0.021*** (0.00) | 0.0012 (0.00) | -0.0078*** (0.00) | -0.0072*** (0.00) |
| Education level of HH (college and above = 1) [†] | 0.0074*** (0.00) | 0.014*** (0.00) | 0.028*** (0.00) | -0.028*** (0.00) | 0.0034*** (0.00) | -0.012*** (0.00) | -0.013*** (0.00) |
| No. of economically active family members, log | 0.16*** (0.00) | 0.018*** (0.00) | -0.068*** (0.00) | -0.016*** (0.00) | -0.045*** (0.00) | -0.014*** (0.00) | -0.030*** (0.00) |
| Spouse employment (employed = 1) | 0.0083*** (0.00) | 0.0018*** (0.00) | 0.0043*** (0.00) | -0.0045*** (0.00) | -0.0056*** (0.00) | -0.00040 (0.00) | -0.0039*** (0.00) |
| Time index | -0.025*** (0.00) | -0.00076*** (0.00) | 0.019*** (0.00) | 0.0026*** (0.00) | 0.012*** (0.00) | 0.0025*** (0.00) | -0.010*** (0.00) |
| Selection bias (if a household did not consume any food item, yes = 1) | 0.0047*** (0.00) | 0.042*** (0.00) | -0.014*** (0.00) | -0.0076*** (0.00) | 0.011*** (0.00) | -0.018*** (0.00) | -0.019*** (0.00) |
| Constant | -0.78*** (0.01) | 0.048*** (0.01) | 0.69*** (0.01) | 0.20*** (0.01) | 0.37*** (0.00) | 0.045*** (0.00) | 0.43*** (0.01) |

Notes: Standard errors in parentheses.

*, **, and *** denote statistical significance levels at 10, 5, and 1 percent, respectively.

[†] Base level is no education. HH = household head.

women's empowerment in grocery decision-making and demand for high-quality premium rice (aromatic and Jasmine rice) is found in South and Southeast Asian consumers [59].

Third, the role of education in diversifying the food basket of Filipino households is evident. Education is included in the regression with four dummy variables, dividing education into five groups: no education (base), elementary, high school, college undergrad and specialized

**Table 6. Parameter estimates from QUAIDS model for urban Filipino households, 2006–2018.**

| Variables | Rice | Other cereals | Meat | Fish | Dairy products | Fruits and vegetables | Miscellaneous |
|---|---|---|---|---|---|---|---|
| Log of prices | 0.260*** (0.01) | -0.059*** (0.00) | -0.150*** (0.00) | 0.029*** (0.00) | -0.045*** (0.00) | -0.00089 (0.00) | -0.032*** (0.00) |
| | | 0.030*** (0.00) | 0.015*** (0.00) | -0.0067*** (0.00) | 0.0083*** (0.00) | 0.013*** (0.00) | -0.00059 (0.00) |
| | | | 0.094*** (0.00) | -0.027*** (0.00) | 0.065*** (0.00) | -0.014*** (0.00) | 0.017*** (0.00) |
| | | | | -0.060*** (0.00) | 0.029*** (0.00) | 0.0076*** (0.00) | 0.028*** (0.00) |
| | | | | | -0.12*** (0.00) | 0.0053*** (0.00) | 0.062*** (0.00) |
| | | | | | | -0.0071*** (0.00) | -0.0040*** (0.00) |
| | | | | | | | -0.070*** (0.00) |
| Log of expenditure | -0.33*** (0.00) | 0.072*** (0.00) | 0.16*** (0.00) | -0.042*** (0.00) | 0.10*** (0.00) | -0.018*** (0.00) | 0.048*** (0.00) |
| Log of expenditure squared | -0.011*** (0.00) | 0.014*** (0.00) | 0.0027*** (0.00) | -0.0049*** (0.00) | 0.0033*** (0.00) | -0.0032*** (0.00) | -0.00073** (0.00) |
| Residuals of expenditure | 0.19*** (0.00) | 0.024*** (0.00) | -0.089*** (0.00) | 0.023*** (0.00) | -0.068*** (0.00) | -0.0089*** (0.00) | -0.073*** (0.00) |
| Gender of HH (male = 1) | 0.013*** (0.00) | -0.0033*** (0.00) | -0.0057*** (0.00) | 0.0062*** (0.00) | -0.0044*** (0.00) | -0.0047*** (0.00) | -0.00075 (0.00) |
| Age of HH, log | -0.0092*** (0.00) | -0.0084*** (0.00) | -0.0024*** (0.00) | 0.018*** (0.00) | -0.0060*** (0.00) | 0.022*** (0.00) | -0.014*** (0.00) |
| Marital status of HH (married = 1) | 0.026*** (0.00) | 0.0034*** (0.00) | -0.016*** (0.00) | 0.0024*** (0.00) | -0.0070*** (0.00) | 0.0045*** (0.00) | -0.013*** (0.00) |
| Education level of HH (elementary school = 1) | -0.014*** (0.00) | -0.0039*** (0.00) | 0.034*** (0.00) | -0.014*** (0.00) | 0.0028** (0.00) | -0.011*** (0.00) | 0.0059*** (0.00) |
| Education level of HH (high school = 1) [†] | -0.015*** (0.00) | 0.000072 (0.00) | 0.039*** (0.00) | -0.020*** (0.00) | 0.0058*** (0.00) | -0.012*** (0.00) | 0.0027* (0.00) |
| Education level of HH (college = 1) [†] | -0.022*** (0.00) | -0.0018 (0.00) | 0.041*** (0.00) | -0.017*** (0.00) | -0.0016 (0.00) | -0.0027** (0.00) | 0.0040*** (0.00) |
| Education level of HH (university = 1) [†] | -0.0078** (0.00) | 0.0028* (0.00) | 0.039*** (0.00) | -0.026*** (0.00) | 0.000053 (0.00) | -0.0046*** (0.00) | -0.0038** (0.00) |
| No. of economically active family members, log | 0.14*** (0.00) | 0.0065*** (0.00) | -0.057*** (0.00) | 0.0031*** (0.00) | -0.048*** (0.00) | -0.014*** (0.00) | -0.033*** (0.00) |
| Spouse employment (employed = 1) | 0.010*** (0.00) | 0.0016*** (0.00) | -0.0016*** (0.00) | 0.00056 (0.00) | -0.0072*** (0.00) | -0.00023 (0.00) | -0.0034*** (0.00) |
| Time index | -0.023*** (0.00) | -0.00017 (0.00) | 0.018*** (0.00) | 0.0013*** (0.00) | 0.015*** (0.00) | 0.0016*** (0.00) | -0.013*** (0.00) |
| Selection bias (if a household did not consume any food item, yes = 1) | 0.013*** (0.00) | 0.0025*** (0.00) | 0.0030*** (0.00) | -0.0014* (0.00) | 0.014*** (0.00) | -0.022*** (0.00) | -0.0095*** (0.00) |
| Constant | -0.81*** (0.01) | 0.20*** (0.01) | 0.68*** (0.01) | -0.0073 (0.01) | 0.42*** (0.01) | 0.022*** (0.01) | 0.49*** (0.01) |

Notes: Standard errors in parentheses.

*, **, and *** denote statistical significance levels at 10, 5, and 1 percent, respectively.

[†] Base level is no education. HH = household head.

training, and college and higher education. The findings indicate that educated household heads spend more on meat and dairy products than non-educated and less educated household heads. Additionally, urban educated households spend significantly less on rice consumption. These findings are consistent with previous demand studies in Asia that noted that

educated households are likely to have more income and consciousness regarding their requirements for foods rich in animal protein and thus spend more on nutritious food. For instance, Bairagi et al. [11] noted that educated Vietnamese households consume more fish. In Bangladesh, Mottaleb et al. [10] found that if both the household head and spouse are educated, they spend more on food items. Similarly, Khanal et al. [27] noted a significant effect of education on food demand in rural India.

Fourth, the age coefficients in Tables 5 and 6 are negative and significant for cereals (rice and other cereals) and miscellaneous food items. This finding underscores the importance of shifting younger consumer preference away from rice and toward meat, fish, and dairy products. Filipino households also spend more on cereal (rice and other cereals) consumption if more economically active individuals are in a family and the spouse is employed. Marital status plays a vital role in food consumption in the Philippines, consistent with expectations that married couples are more likely to spend a greater proportion of their food budget eating at home [60]. Rural married household heads tend to spend less on animal protein, such as meat, fish, and dairy products, than their non-married household counterparts. However, urban married households spend more on fish and fruits and vegetables.

## Estimated expenditure elasticities

The estimated expenditure (income) elasticities across time and rural-urban landscape are presented in Table 7, revealing that all the expenditure elasticities of demand for food items are positive and statistically significant. Among all food items, the magnitude of the expenditure elasticity for rice is the smallest (0.010–0.122 for all samples), similar to what was found by Lantican et al. [25]. This finding indicates that rice is a normal and necessary good in the Philippines since a price change is less likely to affect its consumption. However, the mean rice expenditure elasticities between rural and urban households are significantly different (0.109 vs. 0.069). This means that a 10% increase in income (expenditure) would result in a 1.09%

**Table 7. Expenditure elasticities for food items across rural-urban landscape and time.**

| Food groups | 2006 | 2009 | 2012 | 2015 | 2018 | 2006–2018 |
|---|---|---|---|---|---|---|
| *Rural* | | | | | | |
| Rice | 0.066 | 0.117 | 0.089 | 0.094 | 0.122 | 0.109 |
| Other cereals | 0.554 | 0.556 | 0.480 | 0.476 | 0.480 | 0.496 |
| Meat | 2.127 | 2.145 | 2.035 | 1.984 | 1.988 | 2.033 |
| Fish | 1.118 | 1.123 | 1.123 | 1.125 | 1.130 | 1.126 |
| Dairy products | 2.352 | 2.291 | 2.325 | 2.211 | 2.001 | 2.138 |
| Fruits and vegetables | 0.990 | 0.999 | 1.009 | 1.013 | 1.019 | 1.011 |
| Miscellaneous | 1.285 | 1.321 | 1.340 | 1.374 | 1.427 | 1.370 |
| *Urban* | | | | | | |
| Rice | 0.01[a] | 0.075 | 0.054 | 0.061 | 0.082 | 0.069 |
| Other cereals | 0.843 | 0.801 | 0.740 | 0.714 | 0.681 | 0.725 |
| Meat | 1.949 | 1.978 | 1.950 | 1.923 | 1.932 | 1.943 |
| Fish | 0.921 | 0.933 | 0.948 | 0.955 | 0.964 | 0.951 |
| Dairy products | 2.589 | 2.457 | 2.456 | 2.265 | 1.986 | 2.182 |
| Fruits and vegetables | 1.023 | 1.031 | 1.041 | 1.047 | 1.054 | 1.045 |
| Miscellaneous | 1.259 | 1.288 | 1.300 | 1.326 | 1.369 | 1.325 |

Notes: Authors' estimation based on VHLSS data and QUAIDS model. All elasticities are statistically significant at the 1% level.

[a] Insignificant.

and 0.07% increase in rice consumption in rural and urban areas, respectively. We have also found that the expenditure elasticities of demand for rice, other cereals, and fish are less than 1.00, suggesting that these three food items are normal goods. Meat, dairy products (e.g., milk), and fruits and vegetables are luxury food items (elasticity >1.0). For a luxury food item, consumer demand is more sensitive to income, meaning that a large proportion of a budget is needed to purchase a luxury food item. In other words, meat, dairy products, and fruits and vegetables are expensive items for Filipinos. The expenditure elasticities of demand for meat and dairy products declined considerably for both rural and urban households during 2006–2018, implying a higher demand for these food items.

The expenditure elasticities for all the commodities across income groups are also estimated and presented in Table 8. The Philippine Statistics Authority (PSA) defines income deciles or classes, dividing the entire sample households into ten groups, and each group comprises 10% of the total sample. The expenditure elasticities vary from –0.630 to 2.958, with the lowest for rice and the highest for dairy products. These estimated elasticities are within the range of previous expenditure elasticities (from –5.00 to 3.50) estimated for food demand in South and Southeast Asia [11, 12, 27, 40, 41, 61]. Table 8 further reveals that rice becomes an inferior good for upper-income groups in rural (8-10th deciles) and urban areas (7-10th deciles), as we find that rice expenditure elasticities for these deciles range from –0.099 to –0.507 and from –0.060 to –0.630, respectively. This implies that a 10% increase in the income (expenditure) of the top 30 percentile in rural households and the top 40 percentile in urban households will be accompanied by a 1.77% and 2.50% decline in their rice consumption, respectively. The trend

**Table 8. Expenditure elasticities across different income groups in the Philippines, 2006–2018.**

| Items | Income decile | | | | | | | | | |
|---|---|---|---|---|---|---|---|---|---|---|
| | 1st decile | 2nd decile | 3rd decile | 4th decile | 5th decile | 6th decile | 7th decile | 8th decile | 9th decile | 10th decile |
| *Annual per capita income* | | | | | | | | | | |
| Rural Filipinos (PHP) | 20,420 | 23,322 | 26,602 | 31,246 | 36,791 | 44,499 | 54,680 | 71,791 | 98,119 | 172,698 |
| Urban Filipinos (PHP) | 24,595 | 28,269 | 32,271 | 37,392 | 42,865 | 50,015 | 60,089 | 74,252 | 99,031 | 177,300 |
| *Rural* | | | | | | | | | | |
| Rice | 0.348 | 0.269 | 0.222 | 0.175 | 0.123 | 0.069 | -0.007[a] | -0.099 | -0.232 | -0.507 |
| Other cereals | 0.507 | 0.503 | 0.499 | 0.495 | 0.493 | 0.491 | 0.493 | 0.496 | 0.502 | 0.512 |
| Meat | 2.675 | 2.339 | 2.213 | 2.121 | 2.046 | 1.984 | 1.921 | 1.864 | 1.805 | 1.731 |
| Fish | 1.142 | 1.134 | 1.130 | 1.128 | 1.126 | 1.124 | 1.121 | 1.120 | 1.118 | 1.116 |
| Dairy products | 2.858 | 2.507 | 2.375 | 2.271 | 2.184 | 2.100 | 2.020 | 1.939 | 1.859 | 1.753 |
| Fruits and vegetables | 1.035 | 1.024 | 1.019 | 1.015 | 1.011 | 1.008 | 1.004 | 1.000 | 0.995 | 0.988 |
| Miscellaneous | 1.451 | 1.411 | 1.396 | 1.383 | 1.371 | 1.361 | 1.350 | 1.339 | 1.328 | 1.312 |
| *Urban* | | | | | | | | | | |
| Rice | 0.333 | 0.242 | 0.190 | 0.138 | 0.082 | 0.023 | -0.060 | -0.163 | -0.313 | -0.630 |
| Other cereals | 0.620 | 0.666 | 0.687 | 0.705 | 0.724 | 0.742 | 0.767 | 0.792 | 0.821 | 0.864 |
| Meat | 2.555 | 2.206 | 2.095 | 2.017 | 1.950 | 1.897 | 1.840 | 1.789 | 1.736 | 1.669 |
| Fish | 0.986 | 0.969 | 0.962 | 0.957 | 0.952 | 0.947 | 0.941 | 0.934 | 0.927 | 0.916 |
| Dairy products | 2.958 | 2.535 | 2.402 | 2.305 | 2.222 | 2.140 | 2.064 | 1.983 | 1.901 | 1.791 |
| Fruits and vegetables | 1.075 | 1.061 | 1.055 | 1.050 | 1.045 | 1.041 | 1.036 | 1.031 | 1.025 | 1.017 |
| Miscellaneous | 1.362 | 1.345 | 1.337 | 1.331 | 1.324 | 1.319 | 1.314 | 1.309 | 1.303 | 1.296 |

Notes: The Philippine Statistics Authority (PSA) defined income decile by ranking the weighted total family of all sample families from the lowest to the highest. Then, this is divided into ten groups. The first tenth is called the first decile, meaning people with the lowest income; the second tenth is called the second decile, and so on. All elasticities are statistically significant, at least at the 5% level.

[a] Insignificant.

of rice as an inferior good for higher-income groups is similar to what Mottaleb et al. [10] found in Bangladesh and Bairagi et al. [11] found in Vietnam. The consumption of other food items such as meat, fish, dairy products, fruits and vegetables, and miscellaneous food is expected to rise as income increases since expenditure elasticities are on the decline for these food items. Finally, the results further indicate that the overall demand for foods is likely to be less elastic at higher income levels and for urban households, consistent with previous studies in Vietnam [11, 12] and Bangladesh [10].

## Estimated own- and cross-price elasticities

The compensated (Hicksian) and uncompensated (Marshallian) own- and cross-price elasticities for urban and rural households in the Philippines are presented in Tables 9 and 10, respectively. Compensated price elasticity assumes that consumers are compensated for price changes through budget changes, whereas uncompensated price elasticity assumes that demand changes with price changes, holding the budget constant. Thus, theoretically, compensated elasticities are lower than uncompensated elasticities [62]. Our findings illustrate that the estimated own-price elasticities for all food items are negative and consistent with the economic theory of demand. For example, compensated and uncompensated own-price elasticities for rural households ranged from –0.684 to –3.006 and from –0.726 to 3.151, respectively, whereas for urban households, these elasticities ranged from –0.696 to –3.244 and from –0.760 to –3.390, respectively. The lowest and highest own-price elasticity (absolute) are found for other cereals and dairy products among all of the food items, respectively. For rice, the compensated own-price elasticity is –0.903 for rural households and –0.915 for urban households, indicating that a 1.0% rise in rice price will shrink rice consumption by nearly 1.0% in the Philippines. This estimated price elasticity is within the range of rice price elasticities estimated for Chinese households, from –0.07 to –1.69 [40]. However, it is comparatively higher than what was estimated for Vietnamese households, from –0.24 to –0.60 [11, 12].

The cross-price elasticity of demand reflects the responsiveness of the quantity demanded of a particular food item to a change in the prices of other food items. If the cross-price elasticity of demand between two food items is positive (negative), the goods are called substitute (complement), and a high cross-price elasticity suggests a greater shift in consumer purchases as the price changes. Therefore, these elasticities are useful instruments for policy framing because regulating one food item could affect the demand for other food items not regulated [63]. For example, rice and cereals are substitutes and all other food items are complements, indicating that price-reducing policies for nutritious foods will decrease dependency on the Philippines' rice sector (Tables 9 and 10). The results further reveal that more than 95% of the cross-price elasticities are statistically significant, and the signs of these elasticities vary among food items. In other words, there is a mixture of gross substitutes and complements among food items, meaning that changes in expenditure (income) markedly influence food baskets in the Philippines. A similar finding was also noted for Vietnamese [11] and Indian food baskets [27].

## Conclusions and policy implications

Despite the prevailing poverty and food security challenges, the Philippines has one of the fastest-growing economies in East Asia. Even though the service sector is the main sector contributing to the national GDP, the agricultural sector employs more than one-third of the country's labor force. The main constraints of the agricultural sector are low productivity, limited diversification, and climate change and its impacts. Therefore, the country has historically been a food import-dependent country. To attain self-sufficiency in rice production, the

**Table 9. Estimated own- and cross-price elasticities for food items consumed at home by rural Filipinos (2006–2018).**

|  | Rice | Other cereals | Meat | Fish | Dairy products | Fruits and vegetables | Miscellaneous |
|---|---|---|---|---|---|---|---|
| *Uncompensated* |  |  |  |  |  |  |  |
| Rice | -0.903*** | -0.098*** | 0.198*** | 0.205*** | 0.188*** | 0.079*** | 0.223*** |
|  | (0.005) | (0.006) | (0.006) | (0.006) | (0.007) | (0.009) | (0.007) |
| Other cereals | -0.447*** | -0.726*** | 0.067*** | 0.061*** | 0.163*** | 0.294*** | 0.091*** |
|  | (0.010) | (0.010) | (0.011) | (0.011) | (0.012) | (0.015) | (0.012) |
| Meat | -0.141*** | -0.088*** | -1.050*** | -0.287*** | -0.027** | -0.205*** | -0.235*** |
|  | (0.006) | (0.007) | (0.007) | (0.007) | (0.008) | (0.010) | (0.008) |
| Fish | 0.086*** | -0.020*** | -0.129*** | -1.595*** | 0.257*** | 0.084*** | 0.190*** |
|  | (0.005) | (0.005) | (0.005) | (0.005) | (0.006) | (0.008) | (0.006) |
| Dairy products | 0.209*** | 0.063*** | -0.069*** | 0.430*** | -3.151*** | 0.111*** | 0.269*** |
|  | (0.008) | (0.008) | (0.009) | (0.009) | (0.012) | (0.013) | (0.010) |
| Fruits and vegetables | -0.062*** | 0.172*** | -0.106*** | 0.132*** | 0.143*** | -1.181*** | -0.109*** |
|  | (0.005) | (0.005) | (0.005) | (0.005) | (0.006) | (0.008) | (0.006) |
| Miscellaneous | 0.056*** | -0.023*** | -0.121*** | 0.156*** | 0.173*** | -0.123*** | -1.488*** |
|  | (0.004) | (0.004) | (0.005) | (0.004) | (0.005) | (0.006) | (0.003) |
| *Compensated* |  |  |  |  |  |  |  |
| Rice | -0.872*** | -0.089*** | 0.213*** | 0.222*** | 0.196*** | 0.091*** | 0.239*** |
|  | (0.006) | (0.006) | (0.006) | (0.006) | (0.007) | (0.008) | (0.006) |
| Other cereals | -0.305*** | -0.684*** | 0.136*** | 0.138*** | 0.197*** | 0.351*** | 0.167*** |
|  | (0.010) | (0.010) | (0.010) | (0.011) | (0.013) | (0.015) | (0.012) |
| Meat | 0.442*** | 0.082*** | -0.769*** | 0.030*** | 0.112*** | 0.028** | 0.075*** |
|  | (0.007) | (0.007) | (0.007) | (0.008) | (0.009) | (0.010) | (0.008) |
| Fish | 0.409*** | 0.074*** | 0.027*** | -1.420*** | 0.334*** | 0.213*** | 0.362*** |
|  | (0.005) | (0.005) | (0.005) | (0.006) | (0.006) | (0.007) | (0.006) |
| Dairy products | 0.823*** | 0.242*** | 0.227*** | 0.763*** | -3.006*** | 0.356*** | 0.596*** |
|  | (0.008) | (0.008) | (0.008) | (0.009) | (0.012) | (0.012) | (0.009) |
| Fruits and vegetables | 0.228*** | 0.257*** | 0.034*** | 0.289*** | 0.212*** | -1.065*** | 0.045*** |
|  | (0.005) | (0.005) | (0.005) | (0.006) | (0.006) | (0.007) | (0.006) |
| Miscellaneous | 0.449*** | 0.092*** | 0.068*** | 0.369*** | 0.266*** | 0.034*** | -1.279*** |
|  | (0.004) | (0.004) | (0.004) | (0.005) | (0.005) | (0.006) | (0.003) |

Notes: Standard errors in parentheses.

*, **, and *** denote statistical significance levels at 10, 5, and 1 percent, respectively.

country's staple food, the government of the Philippines has been initiating various production-enhancing policies, such as import tariffs and input subsidies. On the demand side, various price-stabilizing measures have also been in place to keep the price of rice at an affordable level. However, the recent changes in population demographics (e.g., a prominent young working-age group), income growth, and urbanization are likely to transform the food demand structure in the Philippines. Therefore, it is vital to investigate the evolution and diversifying patterns of food consumption for Filipino households.

Several conclusions can be drawn from the current study. First, the staple food, rice, is a normal and necessary good for most consumers in the Philippines, and even more so for rural households. Therefore, the self-sufficiency policy, including rice import tariffs, procurement, and production subsidies, may continue to affect consumers for the foreseeable future to make rice more affordable for poor and marginalized people, especially those in rural areas [18]. At the same time, policy tools that consider the changing consumption behavior of upper-income

**Table 10. Estimated own- and cross-price elasticities for food items consumed at home by urban Filipinos (2006–2018).**

|  | Rice | Other cereals | Meat | Fish | Dairy products | Fruits and vegetables | Miscellaneous |
|---|---|---|---|---|---|---|---|
| *Uncompensated* | | | | | | | |
| Rice | -0.915*** | -0.020** | 0.176*** | 0.144*** | 0.249*** | 0.070*** | 0.228*** |
|  | (0.006) | (0.006) | (0.007) | (0.006) | (0.008) | (0.010) | (0.006) |
| Other cereals | -0.235*** | -0.760*** | 0.049*** | 0.023** | 0.001 | 0.202*** | -0.005 |
|  | (0.008) | (0.008) | (0.010) | (0.008) | (0.010) | (0.014) | (0.009) |
| Meat | -0.192*** | -0.080*** | -1.079*** | -0.228*** | 0.032*** | -0.168*** | -0.227*** |
|  | (0.008) | (0.008) | (0.009) | (0.008) | (0.010) | (0.013) | (0.008) |
| Fish | 0.021*** | -0.006 | -0.082*** | -1.416*** | 0.259*** | 0.048*** | 0.225*** |
|  | (0.006) | (0.006) | (0.007) | (0.006) | (0.007) | (0.009) | (0.006) |
| Dairy products | 0.432*** | -0.128*** | 0.037** | 0.394*** | -3.390*** | -0.014 | 0.487*** |
|  | (0.012) | (0.012) | (0.014) | (0.011) | (0.015) | (0.019) | (0.011) |
| Fruits and vegetables | -0.093*** | 0.134*** | -0.094*** | 0.050*** | 0.068*** | -1.075*** | -0.034*** |
|  | (0.006) | (0.006) | (0.007) | (0.006) | (0.007) | (0.010) | (0.006) |
| Miscellaneous | 0.034*** | -0.056*** | -0.115*** | 0.147*** | 0.254*** | -0.054*** | -1.535*** |
| *Compensated* | | | | | | | |
| Rice | -0.896*** | -0.014* | 0.186*** | 0.154*** | 0.253*** | 0.077*** | 0.240*** |
|  | (0.007) | (0.006) | (0.007) | (0.006) | (0.008) | (0.010) | (0.006) |
| Other cereals | -0.041*** | -0.696*** | 0.159*** | 0.132*** | 0.050*** | 0.282*** | 0.114*** |
|  | (0.009) | (0.008) | (0.009) | (0.008) | (0.011) | (0.013) | (0.009) |
| Meat | 0.327*** | 0.093*** | -0.784*** | 0.062*** | 0.162*** | 0.047*** | 0.094*** |
|  | (0.008) | (0.008) | (0.008) | (0.008) | (0.010) | (0.012) | (0.008) |
| Fish | 0.276*** | 0.078*** | 0.063*** | -1.274*** | 0.322*** | 0.153*** | 0.382*** |
|  | (0.006) | (0.006) | (0.006) | (0.006) | (0.007) | (0.009) | (0.006) |
| Dairy products | 1.015*** | 0.066*** | 0.368*** | 0.720*** | -3.244*** | 0.228*** | 0.847*** |
|  | (0.013) | (0.011) | (0.013) | (0.012) | (0.015) | (0.018) | (0.012) |
| Fruits and vegetables | 0.186*** | 0.227*** | 0.064*** | 0.206*** | 0.137*** | -0.959*** | 0.138*** |
|  | (0.006) | (0.006) | (0.006) | (0.006) | (0.007) | (0.009) | (0.006) |
| Miscellaneous | 0.388*** | 0.062*** | 0.086*** | 0.345*** | 0.342*** | 0.093*** | -1.316*** |

Notes: Standard errors in parentheses.

*, **, and *** denote statistical significance levels at 10, 5, and 1 percent, respectively.

groups by locality (rural and urban) may be vital as our results show that rice happens to be an inferior good for them. This means that, as income grows, these higher-income households will eventually start consuming less rice and more of other food products, particularly meat and dairy products (e.g., eggs and milk and milk products).

Second, consistent with previous studies, education plays a critical role in diversifying the food basket in the Philippines, as we find that educated household heads spend more on meat and dairy products than their non-educated and less educated counterparts. Additionally, gender is identified as a primary driver in food demand, which suggests that consumption of animal proteins will be affected by more active women decision-makers, especially household heads that have greater access to productive and financial resources [64–67]. Younger household heads are likely to shift their food preferences away from rice and toward meat, fish, and dairy products.

Third, for a luxury food item (with expenditure elasticity greater than 1.0), consumer demand is more sensitive to income (expenditure), meaning that a larger proportion of a consumer budget is needed to purchase a luxury food item. In other words, meat, dairy products,

and fruits and vegetables are expensive items for Filipinos. However, the expenditure elasticities of meat demand had been declining for both rural and urban households during 2006–2018. This trend may continue in the future and may thus imply more expected demand for these items. Interventions in the non-rice sector, besides the rice sector, could help supply highly nutritious food items at affordable prices to various groups in the Philippines.

Finally, rice is found to be a substitute food item for other cereals and a complement for all other commodities for both rural and urban Filipinos, suggesting that a meal is incomplete without rice. This also suggests a strong dependence on the country's rice sector. Therefore, policies could be designed to increase the production of high-value products, such as meat, fish, and dairy products, resulting in less dependence on the rice sector. In other words, more diversification in the country's agriculture will result in a sustainable food supply in the future. To this end, although the present study uses a rich dataset and a robust demand model, a few limitations are worth mentioning. For instance, we have used Stone Lewbel price indices because of the unavailability of actual commodity prices, which may not precisely reflect the actual market prices. The current study also does not account for the demand for food items at more disaggregated levels, such as within meat (pork versus chicken) and dairy products (eggs versus milk). Therefore, future research should focus on estimating food demand at more disaggregated levels of food items with actual market prices. Finally, since we have found that female-headed and younger households can play an essential role in future demand for animal proteins, more research is needed to reshape the food basket in the Philippines along these lines.

## Supporting information

**S1 Table. Sub-group commodity budget shares, rural households.**
(DOCX)

**S2 Table. Sub-group commodity budget shares, urban households.**
(DOCX)

## Acknowledgments

We thank the editor, László Vasa, and two reviewers, including Katalin Lipták, for their constructive comments that helped improve the manuscript. The findings and conclusions in this article are those of the authors and should not be construed to represent any official USDA or U.S. Government determination or policy. All errors and omissions are the responsibility of the authors. The authors declare no conflict of interest.

## Author Contributions

**Conceptualization:** Subir Bairagi, Yacob Zereyesus.

**Data curation:** Subir Bairagi, Sampriti Baruah.

**Formal analysis:** Subir Bairagi.

**Funding acquisition:** Samarendu Mohanty.

**Investigation:** Subir Bairagi.

**Methodology:** Subir Bairagi.

**Project administration:** Sampriti Baruah, Samarendu Mohanty.

**Resources:** Samarendu Mohanty.

**Software:** Subir Bairagi.

**Validation:** Subir Bairagi, Yacob Zereyesus.

**Visualization:** Subir Bairagi, Yacob Zereyesus.

**Writing – original draft:** Subir Bairagi, Yacob Zereyesus.

**Writing – review & editing:** Subir Bairagi, Yacob Zereyesus.

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
