## [Decision Letter · Decision Letter 0]

19 Jul 2021

PONE-D-21-19841

Structural Shifts in Food Basket Composition of Rural and Urban Philippines: Implications for Future Food Supply System

PLOS ONE

Dear Dr. Subir Bairagi,

Thank you for submitting your manuscript to PLOS ONE. After careful consideration, we feel that it has merit but does not fully meet PLOS ONE’s publication criteria as it currently stands. Therefore, we invite you to submit a revised version of the manuscript that addresses the points raised during the review process.

We look forward to receiving your revised manuscript.

Kind regards,

László Vasa, PhD

Academic Editor

PLOS ONE

Journal Requirements:

Reviewers' comments:

Reviewer's Responses to Questions

**Comments to the Author**

1. Is the manuscript technically sound, and do the data support the conclusions?

Reviewer #1: Partly

Reviewer #2: Yes

2. Has the statistical analysis been performed appropriately and rigorously? 

Reviewer #1: Yes

Reviewer #2: Yes

3. Have the authors made all data underlying the findings in their manuscript fully available?

Reviewer #1: No

Reviewer #2: Yes

4. Is the manuscript presented in an intelligible fashion and written in standard English?

Reviewer #1: Yes

Reviewer #2: Yes

5. Review Comments to the Author

Reviewer #1: I liked the structure of the manuscript, the coherence of the writing, as well as the design of the statistical methodology.

However, my major concern is that the time series are rather outdated. Statistical Tables on Family Income and Expenditure Survey (FIES) from the final results of the 2018 FIES are also available from March 2020. Still, your research covers data as of the year 2015 the latest. Therefore, your policy recommendations are less meaningful. I would recommend to include the 2018 survey data as well.

Similarly, the literature review should include more papers from recent years (I found only 5-6 ones from the previous 3 years' coverage).In my opinion to validate your results you need to be more updated.

Reviewer #2: The paper is a very nice work!

Please add some methodological part to the abstract and write to main aim of the paper! It is missing now.

Line 32-33. More recent data would be needed. Are they available? If yes, please add to the paper.

It is missing the classical literature review chapter, please modify the introduction chapter and create a literature review chapter.

The methodology and the discussion part are very well introduces and the main results are really valuable and novel!

6. PLOS authors have the option to publish the peer review history of their article (what does this mean?). If published, this will include your full peer review and any attached files.

Reviewer #1: No

Reviewer #2: **Yes: **Katalin Liptak

---

## [Author Response · Author response to Decision Letter 0]

1 Oct 2021

Reviewer #1: I liked the structure of the manuscript, the coherence of the writing, as well as the design of the statistical methodology.

Thank you for the positive appreciation of our paper and the constructive feedback. We have formulated our point-to-point responses to the comments in colored text.

However, my major concern is that the time series are rather outdated. Statistical Tables on Family Income and Expenditure Survey (FIES) from the final results of the 2018 FIES are also available from March 2020. Still, your research covers data as of the year 2015 the latest. Therefore, your policy recommendations are less meaningful. I would recommend to include the 2018 survey data as well.

Your intuition is correct that policy recommendations are less meaningful with outdated data. We are fortunate to receive the FIES 2018 data from the Philippines Statistical Authority with a short period of time; otherwise, we may have to abandon this project. 

In the revised analysis, we have added the 2018 data, and our estimates of elasticities have changed, as expected. However, the main findings remained the same as before. 

Similarly, the literature review should include more papers from recent years (I found only 5-6 ones from the previous 3 years' coverage). In my opinion to validate your results you need to be more updated.

Thank you. As also suggested by the other reviewer, in the revised manuscript, a separate section is entirely dedicated to a literature review on the food demand evolution with more recent papers (please see lines 101-164). However, if you think we have missed some vital food demand studies, please let us know. We are ready to address your further comments.

The following new references have been added:

Colen, L., Melo, P. C., Abdul-Salam, Y., Roberts, D., Mary, S., & Gomez Y Paloma, S. (2018). Income Elasticities for Food, Calories and Nutrients Across Africa: A Meta-Analysis. Food Policy, 77, 116–132. https://doi.org/10.1016/j.foodpol.2018.04.002

Cornelsen, L., Green, R., Turner, R., Dangour, A. D., Shankar, B., Mazzocchi, M., & Smith, R. D. (2015). What happens to patterns of food consumption when food prices change? Evidence from a systematic review and meta-analysis of food price elasticities globally. Health Economics, 24(11), 1548–1559. https://doi.org/10.1002/hec

Femenia, F. (2019). A Meta-Analysis of the Price and Income Elasticities of Food Demand (N 19-03; SMART – LERECO).

Fukase, E., & Martin, W. (2020). Economic growth, convergence, and world food demand and supply. World Development, 132, 104954. https://doi.org/10.1016/j.worlddev.2020.104954

Green, R., Cornelsen, L., Dangour, A. D., Honorary, R. T., Shankar, B., Mazzocchi, M., & Smith, R. D. (2013). The effect of rising food prices on food consumption:systematic review with meta-regression. BMJ (Online), 347(7915), 1–9. https://doi.org/10.1136/bmj.f3703

Hussein, M., Law, C., & Fraser, I. (2021). An analysis of food demand in a fragile and insecure country: Somalia as a case study. Food Policy, 101(April), 102092. https://doi.org/10.1016/j.foodpol.2021.102092

Korir, L., Rizov, M., & Ruto, E. (2020). Food security in Kenya: Insights from a household food demand model. Economic Modelling, 92(July), 99–108. https://doi.org/10.1016/j.econmod.2020.07.015

Lokuge, M. N., Zivkovic, S., Lange, K., & Chidmi, B. (2019). Estimation of a censored food demand system and nutrient elasticities: A cross-sectional analysis of Sri Lanka. International Food and Agribusiness Management Review, 22(5), 717–729. https://doi.org/10.22434/IFAMR2019.0031

Ogundari, K., & Abdulai, A. (2013). Examining the Heterogeneity in Calorie-Income Elasticities: A Meta-Analysis. Food Policy, 40, 119–128. https://doi.org/10.1016/j.foodpol.2013.03.001

Reviewer #2: The paper is a very nice work!

We appreciate the time you spent reviewing our paper. Thank you for the positive appreciation of our paper and the constructive feedback. We have formulated our point-to-point responses to the comments in colored text.

1. Please add some methodological part to the abstract and write to main aim of the paper! It is missing now.

As you suggested, we have revised the abstract, which reads as:

“Price and expenditure elasticities are estimated for seven food categories for rural and urban Filipino households, using Stone–Lewbel (SL) price indices and the quadratic almost-ideal demand system (QUAIDS) model. We used multiple years (2006, 2009, 2012, 2015, and 2018) of the Philippines Family Income and Expenditure Survey (FIES) to estimate the food demand system. The results show that rice is a normal good for most households, particularly for rural consumers. However, it is an inferior good for the top 30% of rural Filipinos and the top 40% of urban Filipinos. As income increases, such wealthy households tend to replace their rice-dominated diet with nutrient-dense food products. Female-headed households, younger households, and households with educated members consume significantly more animal proteins such as meat and dairy products.” 

2. Line 32-33. More recent data would be needed. Are they available? If yes, please add to the paper.

We managed to get the FIES 2018 data, and the new data were added to the analysis. Finally, the relevant discussion was updated accordingly with the new insights. 

3. It is missing the classical literature review chapter, please modify the introduction chapter and create a literature review chapter.

As you suggested, we have added a separate literature section in the revised manuscript. Please see lines 101-164. 

4. The methodology and the discussion part are very well introduces and the main results are really valuable and novel!

Thank you.

---

## [Decision Letter · Decision Letter 1]

3 Nov 2021

PONE-D-21-19841R1Structural Shifts in Food Basket Composition of Rural and Urban Philippines: Implications for the Future Food Supply SystemPLOS ONE

Dear Dr. Bairagi,

Thank you for submitting your manuscript to PLOS ONE. After careful consideration, we feel that it has merit but does not fully meet PLOS ONE’s publication criteria as it currently stands. Therefore, we invite you to submit a revised version of the manuscript that addresses the points raised during the review process.

We look forward to receiving your revised manuscript.

Kind regards,

László Vasa, PhD

Academic Editor

PLOS ONE

Journal Requirements:

Reviewers' comments:

Reviewer's Responses to Questions

**Comments to the Author**

1. If the authors have adequately addressed your comments raised in a previous round of review and you feel that this manuscript is now acceptable for publication, you may indicate that here to bypass the “Comments to the Author” section, enter your conflict of interest statement in the “Confidential to Editor” section, and submit your "Accept" recommendation.

Reviewer #2: All comments have been addressed

Reviewer #3: (No Response)

2. Is the manuscript technically sound, and do the data support the conclusions?

Reviewer #2: Yes

Reviewer #3: Partly

3. Has the statistical analysis been performed appropriately and rigorously? 

Reviewer #2: Yes

Reviewer #3: Yes

4. Have the authors made all data underlying the findings in their manuscript fully available?

Reviewer #2: Yes

Reviewer #3: Yes

5. Is the manuscript presented in an intelligible fashion and written in standard English?

Reviewer #2: Yes

Reviewer #3: Yes

6. Review Comments to the Author

Reviewer #2: The Authors made a lot of modifications on the paper. They made a new structure of the paper in order to make their aims and the methods much clearer. They added some new literature! They revised the abstract and the new abstract is much more better. I can accept the paper for publishing.

Reviewer #3: The paper is a good piece of food consumption research.

The abstract should be more compact, indicating the goals of the research or the research question as well. However, these are missing from the paper anyway, so in the introduction, the research goals/questions should be indicated.

The literature review is well improved. I recommend extending it with these sources from Central Europe (where, due to the transition period, similar tendencies in food consumption could be observed): https://ince.md/uploads/files/1455801255_maketa-es-2-2012.pdf (Consumption characteristics of branded meat products in Hungary: the value chain approach); https://www.researchgate.net/publication/338431810_Evaluating_consumer_behavior_for_consumption_of_milk_and_cheese_in_Gjilan_Region_Kosovo

It is not clear why the the data chapter is before the methodology description. I think it would be more logical to regard the data chapter as a part of the methodology

I recommend the publication of this research paper after these minor improvements.

7. PLOS authors have the option to publish the peer review history of their article (what does this mean?). If published, this will include your full peer review and any attached files.

Reviewer #2: **Yes: **Katalin Lipták

Reviewer #3: No

---

## [Decision Letter · Decision Letter 2]

3 Feb 2022

Structural Shifts in Food Basket Composition of Rural and Urban Philippines: Implications for the Food Supply System

PONE-D-21-19841R2

Dear Dr. Bairagi,

We’re pleased to inform you that your manuscript has been judged scientifically suitable for publication and will be formally accepted for publication once it meets all outstanding technical requirements.

Kind regards,

László Vasa, PhD

Academic Editor

PLOS ONE

Additional Editor Comments (optional):

Reviewers' comments:

Reviewer's Responses to Questions

**Comments to the Author**

1. If the authors have adequately addressed your comments raised in a previous round of review and you feel that this manuscript is now acceptable for publication, you may indicate that here to bypass the “Comments to the Author” section, enter your conflict of interest statement in the “Confidential to Editor” section, and submit your "Accept" recommendation.

Reviewer #2: All comments have been addressed

Reviewer #3: All comments have been addressed

2. Is the manuscript technically sound, and do the data support the conclusions?

Reviewer #2: Yes

Reviewer #3: Yes

3. Has the statistical analysis been performed appropriately and rigorously? 

Reviewer #2: Yes

Reviewer #3: Yes

4. Have the authors made all data underlying the findings in their manuscript fully available?

Reviewer #2: Yes

Reviewer #3: Yes

5. Is the manuscript presented in an intelligible fashion and written in standard English?

Reviewer #2: Yes

Reviewer #3: Yes

6. Review Comments to the Author

Reviewer #2: The Authors made a lot of modifications on the paper. They made a new structure of the paper in order to make their aims and the methods much clearer. They added some new literature! They revised the abstract and the new abstract is much more better. I can accept the paper for publishing.

Reviewer #3: The authors improved the paper based on the reviewers' comments. The paper is now eligible for publication in Plos One.

7. PLOS authors have the option to publish the peer review history of their article (what does this mean?). If published, this will include your full peer review and any attached files.

Reviewer #2: **Yes: **Katalin Lipták

Reviewer #3: No

---

## [Editor Report · Acceptance letter]

22 Mar 2022

PONE-D-21-19841R2 

Structural Shifts in Food Basket Composition of Rural and Urban Philippines: Implications for the Food Supply System 

Dear Dr. Bairagi:

I'm pleased to inform you that your manuscript has been deemed suitable for publication in PLOS ONE. Congratulations! Your manuscript is now with our production department. 

Kind regards, 

on behalf of

Prof. Dr. László Vasa 

Academic Editor

PLOS ONE